# Recycling and Reutilizing Polymer Waste via Electrospun Micro/Nanofibers: A Review

**DOI:** 10.3390/nano12101663

**Published:** 2022-05-13

**Authors:** Xiuhong Li, Yujie Peng, Yichen Deng, Fangping Ye, Chupeng Zhang, Xinyu Hu, Yong Liu, Daode Zhang

**Affiliations:** 1School of Mechanical Engineering, Hubei University of Technology, Wuhan 430068, China; 20200005@hbut.edu.cn (X.L.); pengyj1024@163.com (Y.P.); denyichen@163.com (Y.D.); yefangping@hbut.edu.cn (F.Y.); hgzdd@126.com (D.Z.); 2Beijing Key Laboratory of Advanced Functional Polymer Composites, College of Materials Science and Engineering, Beijing University of Chemical Technology, Beijing 100029, China

**Keywords:** electrospinning, recycling, reutilizing, plastic waste

## Abstract

The accumulation of plastic waste resulting from the increasing demand for non-degradable plastics has led to a global environmental crisis. The severe environmental and economic drawbacks of inefficient, expensive, and impractical traditional waste disposal methods, such as landfills, incineration, plastic recycling, and energy production, limit the expansion of their applications to solving the plastic waste problem. Finding novel ways to manage the large amount of disposed plastic waste is urgent. Until now, one of the most valuable strategies for the handling of plastic waste has been to reutilize the waste as raw material for the preparation of functional and high-value products. Electrospun micro/nanofibers have drawn much attention in recent years due to their advantages of small diameter, large specific area, and excellent physicochemical features. Thus, electrospinning recycled plastic waste into micro/nanofibers creates diverse opportunities to deal with the environmental issue caused by the growing accumulation of plastic waste. This paper presents a review of recycling and reutilizing polymer waste via electrospinning. Firstly, the advantages of the electrospinning approach to recycling plastic waste are summarized. Then, the studies of electrospun recycled plastic waste are concluded. Finally, the challenges and future perspectives of electrospun recycled plastic waste are provided. In conclusion, this paper aims to provide a comprehensive overview of electrospun recycled plastic waste for researchers to develop further studies.

## 1. Introduction

Synthetic polymers such as polyethylene terephthalate (PET), polystyrene or expanded polystyrene (PS) and polyamide (PA) have been widely applied in various commercial and household applications since the 1940s owing to their fascinating properties of low-cost, light weight, corrosion-resistance, and processability [1]. The conveniences and benefits of plastic products have significantly increased their production, leading to a large amount of disposed plastic in landfills. Due to the low biodegradation of plastic waste, the resulting accumulation of white pollution seen everywhere causes adverse environmental issues not only to human beings but also to wildlife such as birds [2,3]. Therefore, figuring out effective methods to deal with plastic waste is urgent.

Landfills, incineration, plastic recycling, and energy production are the current post-consumer treatments for the disposal of plastic waste, as shown in Table 1 [4,5,6,7,8,9,10]. In addition to recycling, the other three approaches have drawbacks, such as waste of space (landfills), high cost (energy production), and toxic byproducts (incineration), which restricts their further development as treatments for plastic waste. Additionally, the traditional recycling method aims to eliminate plastic waste rather than encourage its beneficial, value-added reuse. Modern recycling is a superior choice for utilizing plastic waste in valuable products in response to the circular economy. This minimizes the accumulation of plastic waste to alleviate environmental concern and realizes reutilization to create economic value.

Therefore, cost-effective recycling technologies and higher-value applications should be developed. Due to their specific chemical and physical properties, such as high surface ratio, high porosity, light weights and small diameters nanomaterials have attracted lots of attention in various fields, including bioengineering, electronic devices, energy storage, etc. [11,12,13,14,15]. Commonly, nanomaterials can be prepared by drawing [16], template synthesis [17], phase separation [18], self-assembly [19], centrifugal spinning [20], and electrospinning [14,21]. Electrospinning is the primary choice for fabricating nanomaterials, especially micro/nanofibers, due to its simplicity, low costs, and efficiency. Furthermore, the materials for electrospinning are various, and polymers are the most used. Therefore, electrospinning is a promising way to reuse polymeric waste, turning plastic waste into higher-value products [22]. Many studies have changed recycled plastic waste into fibers [23,24]. Zander et al. prepared nanofibers from pure and mixed waste, including polyethylene (PET), polystyrene (PS), and polycarbonate (PC) by electrospinning and established that their mechanical performance is better than that of commercial polymers with the same molecular weight [25].

Similarly, Esmaeili and colleagues synthesized PET, PS, and PC nano/microfibers by solution electrospinning and investigated the effect of various operational parameters, especially needle diameter and spinning voltage, on the properties of the obtained fibers [26]. Thus, to offer a platform for reviewing the potential of electrospinning in reutilizing plastic waste, this review mainly discusses the advantages of electrospinning for recycling and reutilizing plastic waste and focuses on the electrospinning studies to date of the main plastic waste types containing PET, PS, and PA. In the final section, the challenges and future perspectives of electrospun recycled plastic waste are addressed.

## 2. Advantages of Electrospinning for Reutilizing Polymeric Waste

Electrospinning is a versatile and straightforward technique for fabricating micro/nanofibers employing electrostatic force, which uses polymer solutions or melts as raw materials. A conventional electrospinning system consists of three major components: a spinneret (supplying polymer sources), a high voltage power supply (providing electric forces), and a collector (collecting micro/nanofibers), as shown in Figure 1 [27]. During the process, the polymer solution or melt forms a “Taylor cone” at the end of the spinneret by continuously increasing the spinning voltage between the spinneret and the collector [28]. As the applied voltage exceeds a certain critical value, a polymer jet is ejected from the “Taylor cone” tip, which then solidifies into micro/nanofibers in the collector. The electrospinning technique possesses several attractive merits: simplicity, low cost, mass production, and wide application. There are three primary advantages of the electrospinning technique when recycling and reutilizing plastic waste, as shown in Figure 2 [29,30,31,32,33,34,35,36,37,38,39].

### 2.1. Diversity of Electrospinning Materials

Electrospinning can recycle and utilize plastic waste owing to the technique’s diversity of polymer sources. Traditional electrospinning can be classified into solution electrospinning [40] and melt electrospinning [41], based on the state of the polymer used. In solution electrospinning, if the polymers can be matched with appropriate solvents, they can be electrospun into micro/nanofibers. Additionally, melt electrospinning is proposed to give a chance for the polymers lacking suitable solvents used in solution electrospinning, which expands the polymer choices used in electrospinning. Thus, the two types of electrospinning methods are complementary, making the method adaptable for a variety of materials. It is obvious that the common raw materials for electrospinning are polymers, including natural polymers, synthetic polymers, and hybrid blends [42,43,44]. Fortunately, most polymeric waste belongs to synthetic polymers such as PET, PS, PA, and so on, which can be easily electrospun into micro/nanofibers with appropriate electrospinning processing conditions [25,45,46]. Furthermore, the use of additives (such as nanoparticles) in the raw materials is feasible, leading to a wider range of material choices for recycling and utilizing plastic waste. The diversity of electrospinning materials is the first advantage of electrospinning for recycling and using polymeric waste.

### 2.2. Variety of Modified Electrospinning Techniques

Electrospinning has attracted lots of attention in a wide range of applications due to its simplicity, flexibility, and applicability. However, some challenges exist in traditional electrospinning, such as the residues of organic/toxic solvents (solution electrospinning), complex systems, large fiber diameters (melt electrospinning), and low production rates. To satisfy the requirements of upscaling electrospinning production, multiple modified electrospinning techniques have been developed to solidify their foundation for expanded nanotechnological applications. To realize mass production of electrospinning, needleless electrospinning [47,48], multi-jet electrospinning [49], multi-hole electrospinning [50], gas-assisted electrospinning [51], and centrifugal electrospinning [52] have been demonstrated. Among these modified electrospinning fabrication methods, the needleless electrospinning systems have evolved into many different configurations based on the moving state of the spinnerets, classified into two types: rotating spinnerets and stationary spinnerets. As for the rotating spinnerets in needleless electrospinning, multiple jets are automatically introduced by the movement and vibration of the spinnerets. In comparison, stationary spinnerets in needleless electrospinning utilize external forces, such as magnetic forces, gravity, and gas bubbles to produce multiple jets [48]. Multi-jet and multi-hole electrospinning apply the exact mechanism to alter one jet to several jets by increasing the number of nozzles. In addition, gas-assisted electrospinning and centrifugal electrospinning add other forces to help the electrospinning process upscale fiber production.

Unlike conventional single nozzle electrospinning, all the above-mentioned modified electrospinning techniques can produce multiple jets simultaneously, leading to mass production of micro/nanofibers. That is to say, various electrospinning techniques create a large number of opportunities to transform plastic waste into valuable products for commercial applications. This is the second advantage of electrospinning techniques for recycling and reutilizing plastic waste.

### 2.3. The Complexity of Electrospinning Structures

It is known that the electrospun fibers are usually nonwoven, which limits their applications in a variety of areas such as bioengineering, electronic components and flexible devices. Therefore, many strategies, including management of the electrospinning process and configurations, have been proposed to manufacture ordered fibers [53]. For example, Xia et al. reported that uniaxially aligned nanofibers were prepared by electrospinning with a collector consisting of two separate pieces of electrically conductive substrate [54]. Similarly, Li et al. also obtained uniaxially aligned electrospun fibers by a different way of using a rotating cylinder collector and two oppositely placed metallic needles with opposite voltages in 2006 [55]. Moreover, magnetic-assisted electrospinning was established to fabricate aligned fibrous arrays by Jiang et al. [56]. Those studies import external forces or apply special collectors to prepare ordered fibers by electrospinning. However, it is still hard to control a single jet during electrospinning, even with those modified methods. Fortunately, direct-writing electrospinning (DWE), combining electrospinning with 3Dprinting, was created and can tune a single electrospun jet to the designed path during the fabrication process. This was first proposed by Kameoka and colleagues [57] to prepare patterned fibers using a low spinning voltage and a small spinning distance. Subsequently, numerous studies have been published to take advantage of this technique to obtain specially designed structures [58,59,60,61]. The appearance of the DWE method offers a novel way of fabricating complicated structures with fabulous functions. Furthermore, an electrospinning approach called coaxial electrospinning is suitable for preparing core-shell structures [62,63,64,65]. According to the above discussion, it can be concluded that electrospinning can produce fibers with lots of structures, such as hollow structures, multilayer structures, porous structures, core/shell structures, and so on, which offers diverse opportunities for plastic waste to be electrospun into different patterns adaptable to a growing number of applications [66,67,68,69,70,71].

## 3. Typical Polymer Waste and Reutilization via Electrospinning

In the second section, the advantages of electrospinning for recycling and reutilizing plastic waste are summarized. It is easy to see that electrospinning is an amazing recycling technique for turning plastic waste into high-value products. Thus, lots of studies focus on preparing plastic waste via electrospinning and are shown in Table 2 [25,72,73,74,75,76,77,78,79,80,81,82,83,84,85,86,87,88,89]. In the following parts, the types of polymer waste reutilized through electrospinning are illustrated in detail.

### 3.1. PET

PET is a common synthetic thermoplastic polyester prepared by polycondensation, which can be reused by heating. This plastic has a low crystallization rate, high melting point, and facile processing. Based on the fabricating process and thermal history, it can exist in the form of amorphous (transparent) and semicrystalline (opaque or white). Owing to its superior physicochemical properties, such as chemical resistance and thermal stability and non-toxicity, PET can be employed in numerous fields, especially as one of the most suitable packaging plastics for storing various liquids, including water, soft drinks and milk, and oil. However, the scarcity of space, accumulation threats to the environment, and proliferation of pathogenic microorganisms make recycling PET inevitable. As mentioned before, the traditional recycling approaches of plastic waste have merits and demerits [90]. One of the best chances for recycling and reutilizing plastic waste such as PET is to recreate them as new functional products with special functions via electrospinning. Electrospinning is a technology that can fabricate fibers with diameters in the micro/nano scale, which are an order of magnitude lower than that of fibers produced by conventional melt-blown spinning [91]. Based on the publications related to recycling and reutilizing PET waste by electrospinning, there are several ways to make the best use of recycled PET.

#### 3.1.1. Directly Electrospun Recycled PET into Micro/Nanofibers for Different Applications

Most of the electrospun recycled PET micro/nanofibers are applied for filtration. For instance, Andersson et al. developed a tough mesomorphic fiber membrane with recycled PET by electrospinning, which shows a good performance for smoke filtration, as shown in Figure 3 [72]. The IR-absorbance of the fiber mat increases in the 3000–2850 cm^−1^ and 3500–3200 cm^−1^ ranges after smoke filtration, indicating that the CH and OH are absorbed by fiber mats, respectively. Additionally, the absorption of hydrocarbons/alcohols increases with decreasing the fiber diameter. Similarly, Nosko et al. found that the electrospun recycled PET fibers with an average diameter of around 95 nm show an enhanced filtration efficiency of particles with a size of about 120 nm, and the fibers exhibit good vapor permeability and breathability, which can be promising filtration media for personal protection [73]. Except for using electrospun recycled PET for filtration, researchers have also studied the effects of electrospinning processing parameters on the filtration efficiency of the electrospun recycled fibers. For example, Aguiar and colleagues used recycled PET to develop microfiber membranes and evaluated their properties as air filter media for nanoparticles by changing the solution concentrations [74]. In the same year, they electrospun the same membrane by adjusting solution concentrations and tuning the needle diameter, electrospinning processing time, and collector rotation speed [75]. All the above publications are about the air filtration of electrospun recycled PET fibers [92]. There are also studies focused on water filtration. Zander et al. prepared recycled PET nanofibers via solution electrospinning which can be employed for water filtration [76]. They tested the performance of the filter by passing through an aqueous solution of latex fluorescent beads ranging from 30 to 2000 nm. The result shows that the filters are effective in removing the beads with diameters of 1 and 2 μm in the solution, but they are ineffective for removing the particles under 500 nm. Similarly, Attila et al. electrospun recycled PET to polymer fibers with diameters ranging from 200 to 600 nm, with potential use in filtration [93]. In addition, electrospun recycled PET can be applied in energy storage [77], antibiotics [94], electromagnetic shielding, [95] and so on. Karashanova et al. demonstrated that electrospun recycled PET on paper and textile materials results in a waterproof coating, possibly used for protective clothing and waterproof paper [78]. The above researches utilized solution electrospinning to treat recycled PET. Moreover, Yazdanshenas et al. used melt electrospinning to prepare nanofibers from PET bottles [79]. Additionally, Militky and colleagues applied melt electrospinning to fabricate PET nanofibers from recycled PET bottles and studied the effects of the processing parameters on the morphology of the fibers [80].

#### 3.1.2. Recycled PET as a Raw Source of Electrospinning for Mechanism Research

Electrospun PET fibers can become functional materials and be used as raw sources to perform mechanism studies. For instance, Khatri and colleagues conducted a kinetics and thermodynamic study on the dyeability of electrospun waste PET fibers and found that the obtained fibers display excellent properties, such as superior colorfastness, strong mechanical strength, low processing temperature, etc. [82]. In another study, they utilized recycled PET to fabricate colored nanofibers through electrospinning and they conducted physicochemical studies of the obtained fibers [81]. Zuburtikudis et al. attempted to adjust the electrospinning variables to tailor the average fiber diameter of electrospun lignin/recycled PET fibers [83]. Ahmed et al. conducted a similar study by using the Taguchi design experiment to obtain the optimum electrospinning experimental conditions for recycled PET [84]. However, Marzouqi and colleagues employed the same materials, lignin/recycled-PET, as electrospun raw materials to study the effect of the obtained fibers’ nanoscale dimensions on the carbonization process, which is significant in optimizing the carbonization process [85].

#### 3.1.3. Combine Electrospinning with Other Techniques to Fabricate Functionalized Fibers

In addition to using electrospinning as a method of recycling, researchers also combined electrospinning with other techniques to obtain functionalized fibers. Zuburtikudis et al. fabricated carbon nanofibers by electrospinning a mix of lignin and recycled PET with a carbonization process [86]. Similarly, Attia et al. fabricated lignin-based fibers from a blend of recycled lignin and PET by electrospinning, which can be used to remove methylene blue dye in water [96]. In another study, Khorram and colleagues prepared adsorptive membranes for chromium removal from wastewater with electrospun recycled PET fibers treated with cold plasma and functionalized with chitosan. The results show that the functionalized chitosan electrospun membranes with the plasma treatment possess the highest adsorption capacity compared with membranes without plasma treatment and neat nanofibrous PET [87]. In the study by Sakai et al., a functional fibrous membrane with recycled PET for oil/water separation was fabricated by a two-step method. Firstly, the recycled PET was electrospun into a fibrous membrane and then the membrane was dip-coated with polydimethylsiloxane (PDMS) [88]. In another study by Li et al., hierarchical porous recycled PET was manufactured into fibers, also via two steps: electrospinning and solvent post-treatment, which showed the high efficiency of aerosols and virus capture [97]. Figure 4 illustrates the whole preparation process of the mentioned fibers in [97].

Furthermore, Siyal and colleagues synthesized electrospun recycled PET nanofibers with excellent conductivity and good mechanical strength by electrospinning and electroless deposition methods [89]. In another study, Sinha-Ray et al. used a novel supersonic solution blowing method consisting of an electrospinning scheme coupled with a supersonic converging-diverging de Laval nozzle connected with an air compressor to prepare recycled PET mats for filtration of PM0.1–2 [98]. Apart from preparing functional fibers, one study by Naguib et al. manufactured core-shell fibrous mats in two steps: fabricating recycled PET to fibers by electrospinning and then coating the polyaniline by chemically polymerization, which can be utilized as a potential supercapacitor device (Figure 5) [99].

#### 3.1.4. Electrospun Recycled PET Incorporated with Additives to Obtain Fibers with Unique Properties

To endow better mechanical properties and unique properties (e.g., filtration efficiency) to the electrospun recycled PET fibers, some groups put additives into the recycled PET during electrospinning [100,101,102,103,104]. One type of additive is polymers. For example, Manuel et al. fabricated composite fibers consisting of recycled PET, purchased polyacrylonitrile (PAN) and styrene via electrospinning and investigated their mechanical properties [101]. The composite presents high values of hardness and elastic modulus, which are 4.5 and 7.5 times those of PAN fiber values, respectively. Petrik et al. reported that electrospun recycled PET fibers modified with 2-(aminomethyl) pyridine could be an adsorber for Cu^2+^ from an aqueous solution [102]. Hu et al. prepared two kinds of nanofibrous membranes for membrane distillation with the recycled PET waste via electrospinning. One was pristine and the other was modified with 1H,1H,2H,2H-perfluorodecyltriethoxysilane (FAS), showing a better endurance performance [103]. In another study by Goh et al., a soft, flexible, fluffy and 3D aerogel was made with waste PET for the water treatment of heavy metals and energy harvesting [104]. The aerogel was chemically modified by coating polydopamine (PDA) and soaking with PEI to improve its efficiency and versatility. As shown in Figure 6, the preparation process and the SEM images of the 3D aerogel are presented.

Apart from using synthetic polymers as additives in the above studies, some researchers also employed natural polymers. Sangermano et al. fabricated a nanofibrous membrane with recycled PET and chitosan for oil/water separation and discussed the effect of chitosan concentrations on separation performance [105]. The PET nanofiber membranes may possess amphiphilic properties after being modified by hydrophilic chitosan. Furthermore, the morphology, the chemical composition and wettability of the membranes are related to the chitosan concentrations in the solutions, which affects the selective separation behavior of the membranes. Alena et al. demonstrated that electrospun PET/silk fibroin–composite fibers can be used as aerosol filtration membranes for dirt, bacteria, and viruses [106]. The other type of additive is nanoparticles. Latifi et al. obtained a nanofiber web with recycled PET and nano iron oxide via electrospinning, which possesses microwave absorption characterization and wettability [107]. Alirezazadeh and colleagues electrospun a micro/nanofibrous core-sheath yarn with recycled PET wrapped by PAN containing dimethyl 5-sodium sulfoisophthalate nanoparticles and examined its wicking behavior for filtration [108]. Vanegas et al. revealed that electrospinning the recycled PET and zinc oxide nanoparticles together could obtain fibers with antibacterial and antifungal properties [94]. In a study by Hou et al., icephobic nanocomposite electrospun membranes from recycled PET modified with SiO_2_ exhibited high electromagnetic shielding efficiency with superhydrophobic and icephobic performance [95].

### 3.2. PS

Expanded (EPS) and extruded (XPS) polystyrene, also named styrofoam, is a thermoplastic polymer used in many areas, including the electronics and packaging industries, because of its versatility, light weight, thermal stability, cleanliness, and low cost [109]. The extensive applications of EPS make it one of the abundant plastic wastes because it degrades at a prolonged rate in the natural environment and can exist for years. In addition, EPS is toxic to organisms and tends to accumulate surrounding mercury compounds, leading to a harsh influence on the environment [110,111]. Therefore, it is also significant to transform recycled EPS into valuable products by electrospinning.

Many publications have demonstrated that recycled PS can be fabricated into functional fibers by the electrospinning technique [112]. Shin et al. conducted a series of studies on recycled EPS via electrospinning [45]. In 2005, they obtained electrospun fibers with recycled EPS and natural solvent, which benefits the environment [113]. Then, Shin et al. mixed micro-glass fibers with the electrospun EPS fibers and found the fibers’ separation efficiency improved for filtration of water-in-oil emulsions [114,115,116]. The applications of electrospun recycled EPS are commonly for filtration. For example, Khairurrijal et al. researched electrospun recycled EPS for air filtration [117,118,119]. In 2018, they synthesized nanofiber membranes from waste high-impact PS using the electrospinning method, which showed a suitable application in air filtration based on the contact angle measurement and air-filtration test [117]. One year later, Khairurrijal et al. electrospun recycled EPS to nanofiber mats with different morphology, including smooth fiber, wrinkled fibers, and beaded fibers with various diameters, suitable for air filtration, with high mechanical strength, ultra-hydrophobic surface, and high-quality factors [118].

In addition, to expand the origins of the recycled EPS, Khairurrijal and colleagues fabricated nanofibrous membranes for air filtration with recycled EPS waste from various sources, including food packaging, EPS craft, instant noodle cups, and electronics packaging by the electrospinning technique [119]. Another study by Demir et al. prepared fibrous mats using recycled PS by electrospinning method, which can be applied to treat protein-based solid contents of body fluid medical waste [120] and is a promising adsorbent for remediation of oily wastewater [121]. In another study, the mass production of stacked styrofoam nanofibers was realized through a multi-nozzle electrospinning system with a drum collector [122].

Furthermore, researchers also electrospun recycled EPS with additives to improve the properties of the obtained fibers. The mentioned authors, Khairurrijal et al., electrospun a composite membrane with recycled styrofoam and TiO_2_, which can be applied to degrade wastewater [123]. In a study by Asmatulu et al., nanocomposite fibers with super-hydrophobicity characters were synthesized by electrospinning recycled EPS foam added with titanium nanoparticles and aluminum microparticles of different proportions, promising candidates for water collection, water filtration, tissue engineering, and composites [124]. The results showed that the contact angle of the as-prepared nanocomposites was 157° and their fog water collection capacity was more than 1.35 L/m^2^. In addition, the cost of the nanocomposite materials was only USD 2.67, which can provide the minimum daily water consumption for a two-member household (6 L). Rahman et al. reported that superhydrophobic–hydrophilic nanocomposite fibers with the same materials were electrospun for atmospheric clean water production, which was inspired by the fog-harvesting capability of Stenocara beetles [124,125]. Dandin et al. prepared nanocomposite fibers by electrospinning recycled PS with multiwall carbon nanotubes (MWCNTs) and NiZn ferrite, providing a way for turning plastic waste into high-value new products for several industrial applications, such as transportation, construction, and energy [126]. In addition, to combine other nanomaterials to fabricate electrospun recycled PS with various functions, the researcher also applied electrospinning and post-treatment techniques to obtain functional recycled PS products. For instance, Jalal et al. prepared PS cation exchange membranes from recycled PS waste packing by electrospinning and post sulfonation reactions, which plays an essential role in hydrogen fuel cells [127].

### 3.3. PA

Polyamide (PA) is a semicrystalline polymer with good physicochemical properties which is often used in the form of fibers for a broad set of fields, including fashion, automotive, electronics, etc. Currently, PA production is continuously growing to satisfy the requirements of applications, resulting in an important proportion of polymer waste [128]. Compared with recycled PET and EPS, there is less research about recycled PA. Janalikova et al. prepared a fibrous antibacterial membrane with recycled PA and monoacylglycerol (MAG) blend by electrospinning, which also showed a filtration potential similar to electrospun PET and PS [46]. Another study by Arenas et al. reported a sustainable nanofibrous sound absorption membrane with recycled PA6 and polyvinyl alcohol (PVA) by needleless electrospinning. The resulting membrane showed a high porosity and airflow resistivity [129].

### 3.4. Other Plastic Waste

In addition to the recycled PET, PS, and PA via electrospinning, there are other plastic wastes such as lactic acid (LA), acrylonitrile butadiene styrene (ABS), polyvinyl chloride (PVC), etc. that can be recycled and reutilized by this technique. Kim et al. utilized a novel lactic-assisted 3D electrospinning to produce a low-density 3D polycaprolactone/lactic acid fibrous mesh (3D-PCLS) with recycled LA waste, which can be applied in bone tissue engineering [130]. Khairurrijal et al. used ABS waste to prepare a nanofiber membrane by electrospinning, which can potentially be employed as air filtration media [131]. They also investigated the critical concentration of PVC waste morphology from particles to fibers by electrospinning [132]. Park et al. collected waste polyvinyl(butyral) (W-PVB) from windshields [133]. They fabricated carbon nanofibers from the composites of W-PVB and natural cellulose by electrospinning, carbonization, and the KOH activation approach, which are beneficial for removing the rhodamine B from water.

## 4. Summary and Outlook

Accumulation of plastic waste in landfills has become one of the most significant environmental issues today. Recycling and reutilizing plastic waste into high-value products like micro/nanomaterial is an efficient and viable strategy. Electrospinning is undoubtedly a superior method to transform plastic waste into functional fibers, foams, and other materials. In this review, the advantages of electrospinning in reusing plastic waste and the present advances in recycling and reutilizing plastic waste via electrospinning were briefly summarized. As a versatile method for reusing plastic waste, electrospinning mainly possesses three merits: (1) diverse choices of materials offer plenty of opportunities to reutilize various types of plastic waste as raw materials for electrospinning; (2) novel electrospinning techniques provide limitless possibilities for turning plastic waste into profitable products; and (3) various electrospun structures make plastic waste adaptable for applying in numerous areas. Regarding the superior benefits of electrospinning for recycling and reutilizing plastic waste, the developments in changing plastic waste into a high-value product via electrospinning were discussed based on the common types of plastic waste, PET, PS, and PA.

In the future, electrospinning will certainly be a promising way to recycle and reutilize plastic waste owing to the advantages mentioned. Although electrospun plastic waste has already been profoundly and extensively studied, there are still possibilities and perspectives that need to be further explored. Future advances are potentially concluded from the following aspects: (1) employment of additives, such as nanofillers, with plastic waste or mixed plastic waste to endow the electrospun composites with special functions; (2) deep research on the optimization of electrospun processing parameters for different types of plastic waste to fabricate high-value products with designed properties; (3) further developments for solution electrospinning with natural, non-toxic solvent, or usage of a solvent-free manner such as melt electrospinning in order to realize green manufacturing; (4) combination of electrospinning with other techniques to prepare micro/nano functional fibers(or membranes) with unique structures for expanded explorations of potential industrial applications; and (5) intensive studies on the mass production of electrospun plastic waste to fulfill commercialization.

## Figures and Tables

**Figure 1 nanomaterials-12-01663-f001:**
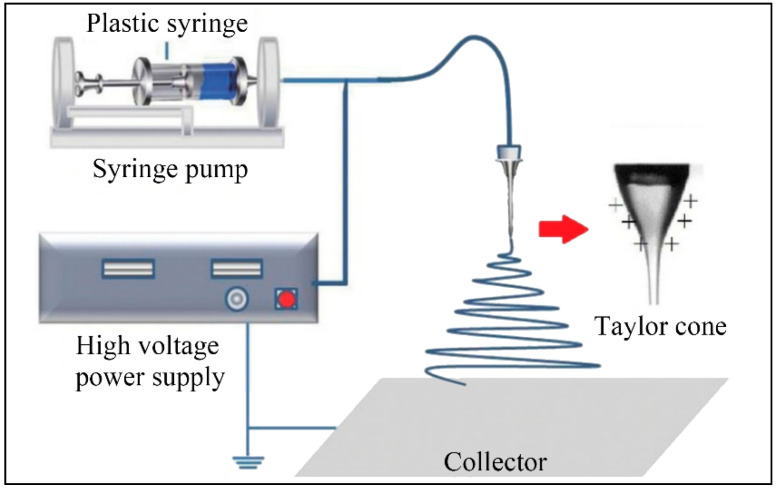
The mechanism of electrospinning. Reprinted from ref. [27].

**Figure 2 nanomaterials-12-01663-f002:**
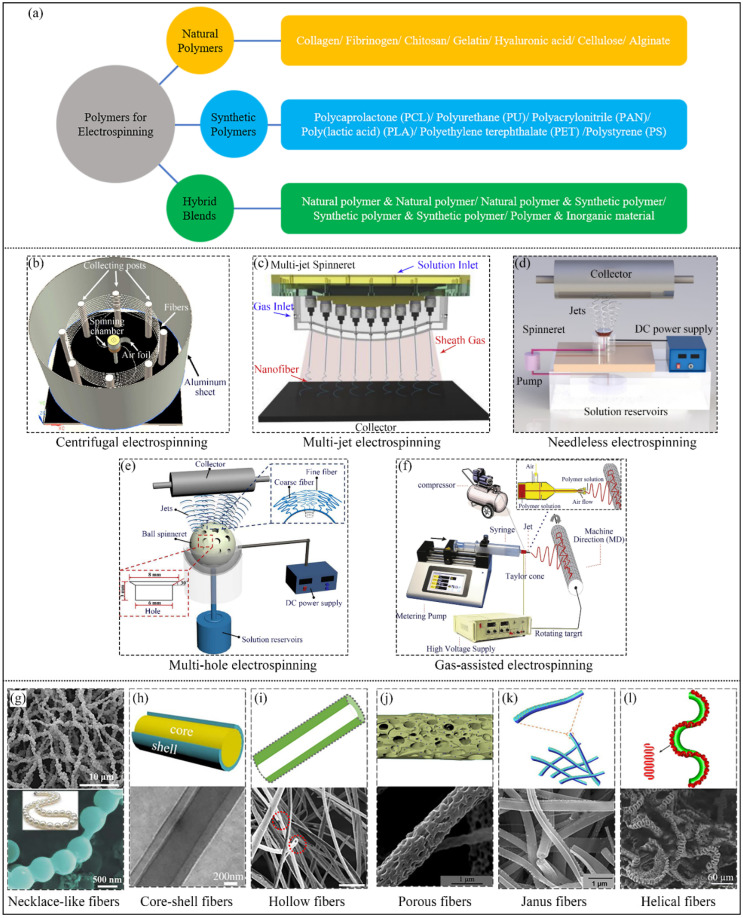
(**a**) Materials for electrospinning (**b**–**f**). Various technologies for electrospinning: (**b**) centrifugal electrospinning [29], (**c**) multi-jet electrospinning [30], (**d**) needleless electrospinning [31], (**e**) multi-hole electrospinning [32], and (**f**) gas-assisted electrospinning [33] (**g**–**l**) Special structures of electrospun fibers: (**g**) necklace-like fibers [34], (**h**) core-shell fibers [35], (**i**) hollow fibers [36], (**j**) porous fibers [37], (**k**) Janus fibers [38]. and (**l**) helical fibers [39]. (**b**,**j**) Reprinted with permission from ref. [29,37]. Copyright American Chemical Society. (**c**–**i**) Reprinted with permission from ref. [30,31,32,33,34,35,36]. Copyright Elsevier. (**k**) Reprinted with permission from ref. [38]. Copyright 2019 Springer. (**l**) Reprinted with permission from ref. [39]. Copyright 2020 Royal Society of Chemistry.

**Figure 3 nanomaterials-12-01663-f003:**
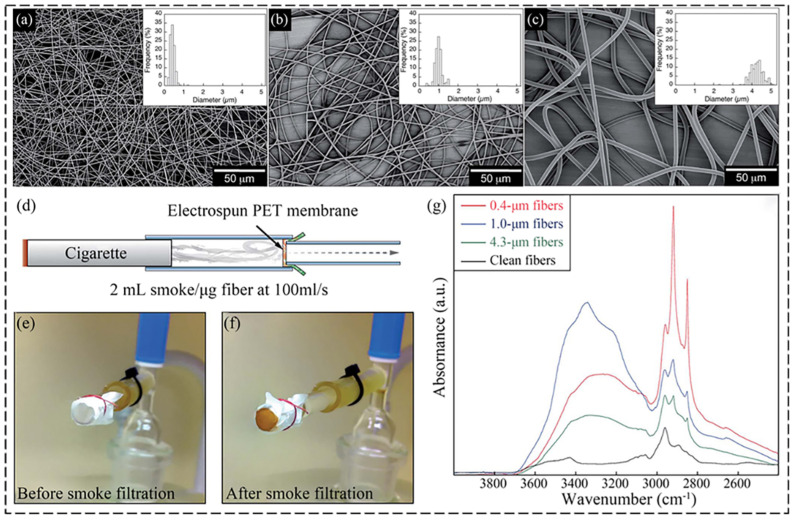
(**a**–**c**) Micrographs of electrospun fibers with different concentrations of PET: (**a**) 10 wt%, (**b**) 15 wt%, (**c**) 20 wt%. (**d**) Schematic illustration of smoke filtration testing (**e**–**f**). Photographs of fiber mats: (**e**) before and (**f**) after smoke filtration testing. (**g**) IR-spectroscopy of clean fiber mats and smoke-exposed fiber mats with average fiber diameters of 0.4, 1.0 and 4.3 mm. Reprinted with permission from ref. [72]. Copyright 2015 Royal Society of Chemistry.

**Figure 4 nanomaterials-12-01663-f004:**
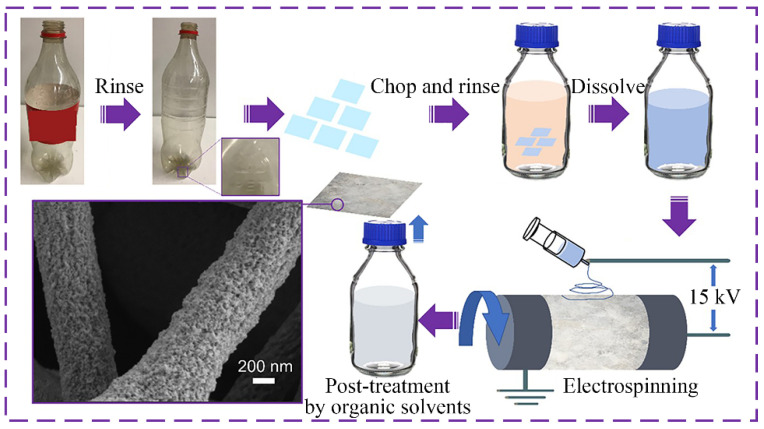
Process of recycled PET electrospinning and post-treatment. Reprinted with permission from ref. [97]. Copyright 2021 American Chemical Society.

**Figure 5 nanomaterials-12-01663-f005:**
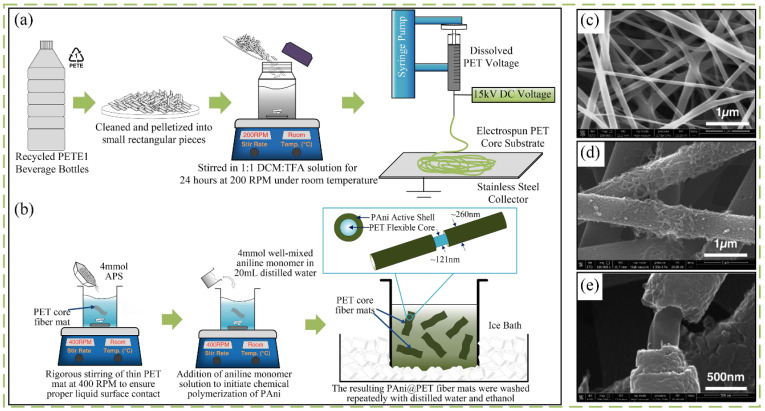
(**a**) Process of fabricating the PET core fibers from the recycled PETE1 recyclable beverage bottles. (**b**) The production process of PAni@PET core-shell fibers. (**c**) Morphology of the pure PET fibrous core. (**d**) PAni@PET core-shell structure. (**e**) SEM photograph of the exposed core. Reprinted with permission from ref. [99]. Copyright 2016 IOP Publishing.

**Figure 6 nanomaterials-12-01663-f006:**
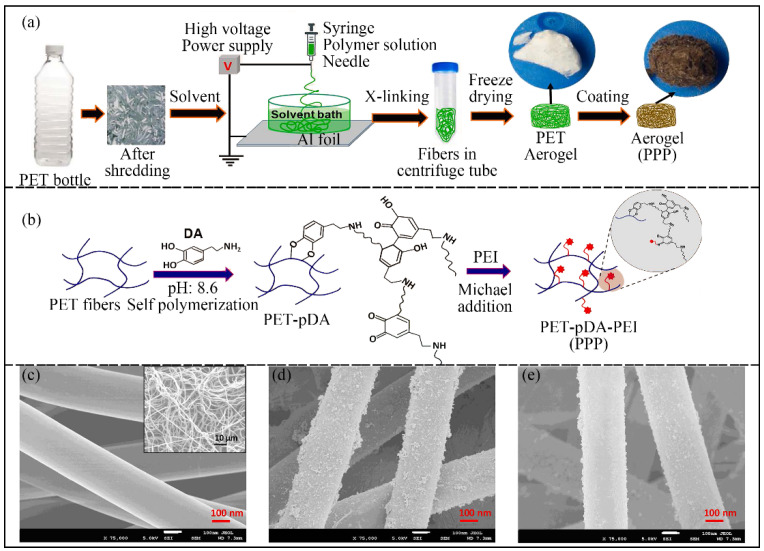
(**a**) Process of fabricating the 3D PPP aerogel from PET bottles. (**b**) Mechanisms for Michael addition reaction occurred during the surface modification process. (**c**–**e**) FESEM images of fibers: (**c**) pristine PET fibers, (**d**) pDA coated fibers (PET-pDA), and (**e**) PET-pDA-PEI (PPP) fibers. Reprinted with permission from ref. [104]. Copyright 2021 Elsevier.

**Table 1 nanomaterials-12-01663-t001:** Various management methods of plastic waste and their influence.

Management	Advantages	Disadvantages	Ref.
Landfill	Convenience	Occupies large space/ Difficult to decompose	[4]
Incineration	Recovery of thermal energy/ Reduction in the weight and volume of waste	Harmful to the environment	[5]
Mechanical pulverization (Physical recycling)	Reused as materials compounded with additives	Remanufactured products with worse properties	[6]
Microbial decomposition	Reduces secondary pollution	Needs special conditions Releases methane gas	[7,8]
Thermal decomposition (Chemical recycling)	Low pollution/ High utilization rate/ High value of products	Relatively complicated	[6,9]
Physical and chemical modification reuse	Recycled products with good properties and values/ Less secondary pollution	Needs high-level techniques and high costs	[6]
Mechanical recycling	Economical/ Environmental	Difficult to recycle complex and contaminated polymer waste/Intense energy consumption	[9,10]

**Table 2 nanomaterials-12-01663-t002:** Summary of polymeric products (membranes or fibers) prepared from plastic waste sources of PET, PS, PA, etc., via electrospinning.

Products	Origin Waste Source	Performance Discussion (Indicator)	Application	Ref.
PET, PS, PC nanofibers	Water bottles, styrofoam, Compact discs (CDs)	Elastic moduli: 15 to 60 MPa/ High water filtration efficiency (over 99%) of 1 μm particles	Ultra/micro filtration	[25]
Tough mesomorphic fiber membranes	Coca Cola bottles (500 mL)	Fiber diameters: 0.4 to 4.3 μm/ High strength (62.5 MPa), modulus (1.39 GPa), toughness (65.5 MJ m^−3^)/ High absorption capacity of smoke residuals (43 × its own weight)	Smoke filtration	[72]
PET nanofibrous membranes	Beverage bottles	Fiber diameters: 95 ± 37 nm/ High filtering efficiency (more than 98% for particles over 120 nm)	Filtration media in face mask	[73]
PET membranes	PET bottles	Fiber diameters: 1.29 μm/ High mechanical resistance (4 MPa)/ High collection efficiency (98.4%) and low-pressure drop (212 Pa)	Air filters	[74]
PET fibers	Clear soda packaging	Mean fiber diameter: 3.25 to 0.65 μm/ Mechanical strength: 3.2 to 4.5 MPa/ High filtration efficiency (up to 99%)	Air/gas filtration	[75]
PET membranes	Plastic water bottles	Fiber diameters: 100 nm/ High filtration efficiency (more than 99% of particles as small as 500 nm/6 log reductions for Gram-negative and Gram-positive bacteria)	Water filtration	[76]
Electrochemical active microporous carbon structure	Used PET bottles	The medium combines double-layer and redox reaction pseudocapacitance characteristics.	Energy storage	[77]
PET films on paper and textile materials	Used mineral water bottles	The impregnated with PET do not absorb water droplets.	Waterproof materials	[78]
PET nanofibers	Used grade PET bottles, PET granule	Minimum fiber diameters: 61 to 93 nm (Produced by melt-electrospinning)	-	[79]
PET nanofibers	Clear PET bottles	Fiber diameters: 45 to 65 μm	-	[80]
Colored PET nanofibers	Recycled PET bottles	Good colorfastness/ Good mechanical strength	Advanced colorful applications	[81]
PET nanofiber mats	Recycled PET bottles	Good colorfastness/ Good mechanical strength/ Low processing temperature/ Minimum dyeing time	Advanced apparel applications	[82]
PET/lignin nanofibers	Waste water bottles	Average fiber diameter: 191 ± 60 nm	Separators/ filters	[83]
Sooth uniform PET nanofibers	Waste PET materials	Minimum average fiber diameter: 105.03 ± 36.79 nm	-	[84]
Lignin/recycled PET fibrous mats	Waste water bottles	Average fiber diameter: 80 to 781 nm	-	[85]
Carbon nanofibers	Used PET water bottles	Average fiber diameter: 191 ± 60 nm/ The C content of the nanofibers: 94.3%	Advanced separation	[86]
Adsorptive membranes	PET bottle waste	Cr(VI) removal capacity (5.54 mg/50 mg)/ Reusability (93.7% adsorption effectiveness after five cycles)	Removal of hexavalent chromium from water	[87]
PDMS functionalized PET fibrous membranes	Recycled PET pellets	Superoleophilic properties (oil contact angle of 0°)/ Anti-water-fouling properties/ High flux (~20,000 L m^−2^ h^−1^)/ High separation efficiency (>98%)	Oil/water separation	[88]
Conductive PET nanofibers	Water bottles	Average fiber diameter of copper-coated r-PET nanofibers (700 nm)/ Low electrical resistance (0.1 Ω)/ Flexible/ Good mechanical strength	Wearable electronics/ Flexible sensors/Energy storage	[89]

## Data Availability

Not applicable.

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
