# Peer review of "Recycling and Reutilizing Polymer Waste via Electrospun Micro/Nanofibers: A Review"

_nanomaterials, 2022, doi:10.3390/nano12101663_

Round 1

Reviewer 1 Report

-In Introduction part, the novelty should be highlighted. How does this review stand out from the existing ones? What's new?

-There is no comments in the text for Table 2.

-Line 122 “In comparison, (…)” should be moved to line 121.

-There is no comments in the text for Figure 2.

-The IR-spectroscopy data on Figure 3g should be commented in the text. Please explain the reason of the changes in the intensities? What compounds are retained on the filter?

-Line 202 should be moved to line 201.

-Lines 199-203, there should be more details about the filtration efficiency, what type of pollutants have been removed.

-Line 275, there should be superscript. Please check this.

-Lines 272-273, what was achieved thanks to this.

-Lines 281-282 please explain in the text why coating polydopamine (PDA) and soaking with PEI have been used to prepare the PET-based aerogel.

-Line 286, why chitosan has been added.

-Lines 288 filtration of what?

-Line 297, there should be subscript.

- Generally in subsection 3.1.4. there is no explanation as to why the various additives (e.g. chitosan or silk fibroin) were added,  and what properties they imparted to these PET materials.

-There is no comments in the text for Figure 6.

-Line 336, similarly to what?

-Line 338, there should be subscript.

-Lines 342-343, there should be more highlighted that this technology is for mas clean water production.

- The authors mention the possibility of using these recycled polymers, e.g. EPS, in tissue engineering. So are there any studies on their cytotoxicity or biocompatibility with cells?

-Line 348, new products for what?

Reviewer 2 Report

The manuscript presents interesting review of the possibilities to form micro/nanofibers from recycled polymers via electrospinning. The information presented by the authors are valuable and scientifically interesting. Manuscript is written in clear style, but some points require additional information.

  • The title of manuscript should be clarified (term “micro/nanofibers” is used in the text, while in the title is written “nanofibers”).
  • There is a lack of logic in providing information. In my opinion, Figure 1 should be presented and described in the same chapter (section) as Fig 2. Besides, Table 2 should be moved to Chapter 3 where polymer waste and their reuse are described.
  • Reference to Table 2 should be presented in the text.

  • Summarizing of the information presented in the Table 2 would be helpful.

Reviewer 3 Report

Title:

The title must indicate the objective for which the research is being carried out (new technique, a problem without a current solution, new study variables, etc.). If the work is a state of knowledge, it must indicate that it is a Review

Abstract:

If the work is a review of knowledge, indicate the limits of the study (time, technology, types of materials, choice of information, etc.) It is advisable to present the most significant conclusions of the work at the end of this section

Keywords:

OK

Introduction:

All the arguments presented must be referenced to the work of other researchers; otherwise, they MUST be demonstrated by the authors. For example: Table 1. Review ALL the document

Table 2: It is necessary for the authors to make an effort to simplify or group the large amount of information presented in this table.

Figure 2: It is necessary for the authors to make an effort to simplify or group the large amount of information presented in this figure.

Figure 3. Check the correct order with the description.

GENERAL NOTE: It is considered that the work presented does not reach an adequate complexity to be considered a review of the subject. Currently it is a "list" of previous works; but it is necessary, by the authors, a deep analysis, contrast of results, exposition of hypothesis and contrast of results. This allows the work to reach a degree of review of the topic. It is necessary that this improvement is made in order to accept the publication of the work.

Bibliography:

Ok

Reviewer 4 Report

The article can be accepted after addressing the following minor comments:

Introduction and Table 1. Please discuss separately the "pros" and "cons" of mechanical (secondary) and chemical (ternary) recycling

Section 2.1 This should include more examples of recycled polymers different than PET and discussed later in section 3

Section 2.2 and 2.3 can be merged. Moreover, the information included in this review can be valuable to extend the information of the manuscript: https://doi.org/10.1002/mame.202100858

The benefits of using electrospinning vs (conventional) melt spinning can be discussed in section 3.1.

In section 3.2., please note PS and EPS (and also XPS) is the same polymer (polystyrene, PS) in the form of a foamed product. The first paragraph is confusing and it is not clear the way it can affect the recylclability or electrospinnability of PS

Authors may include a new section dealing with the potential advantages of the electrospinning process to develop advanced materials with higher added value, such as biomaterials, functional materials, active materials, engineering materials, etc., based on recycled polymers

Round 2

Reviewer 1 Report

Many thanks to the authors for their comprehensive responses to my comments. The paper is ready to be published in the journal.

Reviewer 2 Report

Thanks to the authors for making changes to the manuscript. The overall quality has improved and in my opinion the manuscript can be accepted for publication in Nanomaterials.

Reviewer 3 Report

The requested improvements have been made. The work can be published.